# Effect of Lupin-Enriched Biscuits as Substitute Mid-Meal Snacks on Post-Prandial Interstitial Glucose Excursions in Post-Surgical Hospital Patients with Type 2 Diabetes

**DOI:** 10.3390/nu12051239

**Published:** 2020-04-27

**Authors:** Sophie Skalkos, George Moschonis, Colleen J. Thomas, Joanna McMillan, Antigone Kouris-Blazos

**Affiliations:** 1Department of Dietetics, Nutrition and Sport, School of Allied Health, Human Services and Sport, La Trobe University, Victoria 3086, Australia; foodwiseapd@optusnet.com.au (S.S.); g.moschonis@latrobe.edu.au (G.M.); jo@drjoanna.com.au (J.M.); 2Department of Physiology, Anatomy and Microbiology, School of Life Sciences, La Trobe University, Melbourne, Victoria 3086, Australia; colleen.thomas@latrobe.edu.au

**Keywords:** T2DM post-operative patients, lupin, biscuits, interstitial glucose, continuous glucose monitoring

## Abstract

Hospital biscuit snacks offered to Type 2 Diabetes Mellitus (T2DM) patients may adversely affect glycaemic control. This study investigated the effect of lupin mid-meal biscuit snacks, compared to spelt or standard hospital biscuits, on interstitial glucose levels in post-operative T2DM inpatients. In a pilot cross-over pragmatic study, 20 patients (74 ± 12 years) consumed, in order, lupin biscuits (20% lupin), wholemeal spelt and standard plain sweet biscuits as mid-meal snacks (2 biscuits each for morning and afternoon tea) on three consecutive days. Continuous glucose monitoring, appetite perceptions and bowel motions were recorded. Glucose levels were not significantly different in the first 90 min after mid-meal biscuit consumption at morning and afternoon tea, irrespective of type. However, after consuming the lupin biscuits only, glucose levels were significantly (*p* < 0.001) reduced 90 min postprandially after dinner, indicating a potential second-meal effect. Patients also reported improved satiety after lupin biscuit consumption on day 1, compared to days 2 and 3 (*p* = 0.018). These findings suggest that lupin-enriched biscuits may improve both glycaemic control and satiety in hospitalised T2DM patients, potentially contributing to reduced length of stay. Larger controlled studies are warranted to confirm these findings and inform potential revision of hospital menu standards for T2DM patients.

## 1. Introduction

Type 2 Diabetes Mellitus (T2DM) affects approximately 1.2 million Australians, and its prevalence continues to rise, making it the fastest growing chronic disease in the country [1]. In addition to the known determinants of diabetes, namely obesity, poor dietary habits and sedentary lifestyles, another contributing factor is the rising number of immigrants to Australia from countries with a traditionally higher prevalence of T2DM compared to Australia-born residents [2]. Intervention strategies that promote healthy dietary and lifestyle modifications in T2DM patients are paramount to manage the disease and lower the risk of diabetic complications.

Patients with T2DM are often over-represented in hospital inpatient populations due to diabetic complications [3]. Moreover, patients having surgery with co-morbid T2DM and dysglycaemia (i.e., hyperglycaemia, hypoglycaemia, or excessive variability in glucose levels) are at increased risk of perioperative complications, such as the need for transfusion, pneumonia, delayed discharge, surgical infections and in-hospital mortality [4,5,6,7,8]. Treating hyperglycaemia in the peri-operative period has been shown to reduce complications in several surgical disciplines [4]. Currently, however, there are no uniform guidelines for optimising glycaemic control in T2DM patients undergoing surgery in the pre-and per-operative phases of care. Considering the above, hospital meals and snacks could play an important role in managing dysglycaemia in surgical patients with T2DM. Australian hospital menus are guided by nutrition standards developed by the Australian Government Department of Health [9,10] in conjunction with credentialed dietitians and food service managers. At present, hospital menus typically provide patients with moderate to high amounts of carbohydrates (135–315 g/day). However, a growing body of evidence suggests the potential benefits of a lower carbohydrate diet (i.e., <130 g/day) in the clinical management of T2DM [11,12,13,14,15].

Although the nutrition standards used to structure the hospital full ward menu define the basic food and nutrition needs of adult patients, they do not specify carbohydrate restriction for specialised therapeutic diets for patients with T2DM. Moreover, they do not specify standards for mid-meal snacks (such as biscuits) [16] to provide healthier, fibre-rich and low Glycaemic Index (GI) quality carbohydrate meal options. In the absence of tight regulation, meal and snack offerings can vary significantly between hospital sites depending on the interpretation by hospital catering staff. This usually leads to a homogenised general ward diet that is assigned as suitable for most patients, including those with T2DM. Incorporating healthier mid-meal options on the ward diet could be a practical approach to reduce glucose excursions in patients with T2DM, considering that due to loss of appetite secondary to surgery, patients typically consume a considerable proportion of their daily dietary energy and nutrients intake at mid- meals. 

Lupin is a legume commonly consumed in the Mediterranean diet, and it has a unique nutrition profile. Firstly, it is an excellent source of protein (39 g/100 g; double the protein content of other legumes, with the exception of soy) and provides all the essential amino acids, with a particularly high concentration of arginine [17]. Lupin is also very low in available carbohydrates (only 6 g/100 g), exceptionally rich in fibre (32 g/100 g; double to triple the fibre content of other legumes) and has a very low GI (<10 out of 100) [17]. Furthermore, lupin contains bio-active prebiotic sugars (~8% oligosaccharides) [17]. It has only 8 g/100 g of fat, with minimal saturated fat (1.5 g/100 g) [17]. Finally, lupin is rich in potassium, magnesium and calcium and contains the glycoprotein gamma (γ)-conglutin, not found in any other legume. Gamma (γ)-conglutin has been shown to cross the intestinal barrier, lower blood glucose levels and have anti-inflammatory actions [17].

Lupin flour-enriched foods have shown promise for improving post-prandial glycaemia, but the available evidence from human studies is limited [17]. Legumes and wholegrains may reduce postprandial glycaemia not only at the meal in which they were consumed, but also later in the day, or even the following day, a phenomenon known as a “second-meal effect” [18]. Therefore, the second-meal phenomenon has important implications for the control of day-long blood glucose and may be partly responsible for the reduction in diabetes incidence associated with increased whole grain and legume intake [19]. There are a number of possible mechanisms whereby ingestion of legumes could lower blood glucose levels at a subsequent meal. For example, the effect of immediate reductions in glycaemia (following the initial meal) due to the viscous fibres in legumes is known to slow digestion and absorption [17]. Legumes also provide indigestible carbohydrates for colonic fermentation [17]. The second-meal effect has been reported for various legumes, but it has not been studied in lupins [19,20]. Furthermore, to our knowledge, hospital menus do not presently offer legume-based meals and snacks.

The primary aim of the present pilot pragmatic intervention study was to examine the effect of replacing standard hospital biscuits (low in fibre, high in refined starch and added sugar) routinely offered at mid-meals, with carbohydrate-matched lupin or wholemeal (spelt) biscuits (higher in fibre and protein) on the interstitial glucose levels of hospitalised post-surgical patients with T2DM. A secondary aim was to investigate the effect of lupin enriched biscuits on study participants’ satiety, palatability and bowel motions compared to the other two types of biscuits. 

## 2. Materials and Methods 

### 2.1. Study Design

A controlled cross-over pragmatic study design was followed, whereby all subjects received three different treatments on three consecutive days at morning and afternoon tea, approximately 10 am and 2 pm, respectively. Randomisation was not applied as each subject consumed identical treatment on each consecutive day. As such, each study participant served as his or her own control, thus removing the effect of subject-to-subject variation. The lupin biscuits were provided first since they were expected to have the lowest impact on post-prandial glucose levels and, therefore, less potential for carry-over effect of raised glucose levels. The pilot study was reviewed and approved by the Human Research Ethics Committee of La Trobe University (approval number HEC 17-073), and permission to run the pilot study at Warringal Private Hospital was provided by the Director of Clinical Services. All participants involved provided informed written consent to participate. Finally, the study was registered to the Australian New Zealand Clinical Trials Registry (ACTRN12620000353998).

### 2.2. Study Participants

Twenty-five post-surgical hospital patients with T2DM were recruited for the pilot study. Convenience sampling was undertaken by the researcher conducting the field work, in scope of their role as a Senior Dietitian and Diabetes Educator, using the hospital database to identify and recruit patients with T2DM admitted to Warringal Private Hospital in Melbourne, Victoria, Australia. English-speaking patients aged > 50 years were eligible for study inclusion if they were diagnosed with T2DM, managing T2DM via oral hypoglycaemic agents and admitted for surgery. Patients were excluded from the study if they had known allergies to wheat, eggs, nuts, legumes, corn and/or dairy, diagnosed Irritable Bowel Syndrome (IBS) or followed a low Fermentable Oligosaccharide Disaccharide Monosaccharide and Polyol (FODMAP) diet, diagnosed dementia or mental illness, non-English speaking or currently taking prescribed steroidal or insulin medications. Patients were also given the opportunity to discuss participation in this study with family members and reassured that their hospital care would not be affected if they chose to opt out. 

### 2.3. Intervention

Recruitment commenced May 2018 and concluded December 2018. Post-surgery patients were asked to consume two biscuits at each morning tea and afternoon tea, in addition to a provided 45 g carbohydrate-controlled meal at breakfast, lunch and dinner, for three consecutive days. The rationale for the 45 g quantity of carbohydrates in the main meals was that the hospital did not want to over-restrict patient food choices, and it also wanted to be in line with the moderate carbohydrate intake recommended by the Agency for Clinical Innovation (New South Wales, Australia) for hospital patients with diabetes [9]. On the first day, study participants were provided with four lupin biscuits (Skinnybik™ Lupin, Murrumbeena, Victoria, Australia) containing 20% lupin flour (equivalent to a total of 1 tablespoon of lupin flour), on the second day with four wholemeal spelt biscuits (Skinnybik™ Spelt, Murrumbeena, Victoria, Australia) and on the third day with four hospital standard plain sweet biscuits (Arnott’s Marie Biscuits™, NSW, Australia). Continuous glucose monitoring (CGM) devices were fitted to each patient in the evening before commencing the three-day diet intervention. Participants were instructed to refrain from consuming any extra snacks during the day and to not add sugar to hot beverages.

### 2.4. Nutrient Composition and Ingredients in the Study Biscuits

Table 1 summarises the nutrient composition and ingredients of the study biscuits. Briefly, the lupin and spelt biscuits had identical carbohydrate (36 g per 100 g) and fibre content and similar protein and fat content. Arnott’s biscuits had double the carbohydrate content of the lupin and spelt biscuits (i.e., 75 g per 100 g). Therefore, the portion size of these biscuits was halved to match the carbohydrate amount consumed for mid-meals on the lupin or spelt days and to assist in identifying any unique glycaemic response effect of lupin biscuits compared to the other two biscuits. All biscuits were removed from their packaging, and all had a similar colour (light brown) and appearance so as not to overtly influence the participants regarding their flavour or visual identification.

### 2.5. Socio-Demographic and Medical History Data

Socio-demographic and medical history data of study participants were collected from the hospital medical records. This information included patient age, gender and country of birth (ethnicity), as well as the type and dosage of oral hypoglycaemic agents. Type of surgical procedure and year of initial T2DM diagnosis was also recorded.

### 2.6. Adherence to Biscuit Intervention and Plate Wastage 

The dietitian visited participants at meal and mid-meal times. For the three study days, the amount of food and biscuits consumed by study participants in the main and mid-meals were estimated by the dietitian through visual inspection. The amount of biscuits consumed and plate wastage were taken into account when calculating the amount of carbohydrate consumed by study participants.

### 2.7. Anthropometric Measurements

Anthropometric measurements were conducted at hospital admission. Body weight and height were measured in study participants wearing light clothing and no shoes, using a digital scale (Seca Alpha, Model 770, Hamburg, Germany) with an accuracy of ± 100 g and a commercial stadiometer (Leicester Height Measure, Invicta Plastics Ltd., Oadby, UK) to the nearest 0.5 cm, respectively. Body mass index (BMI) was calculated as weight (kg) divided by height squared (m^2^). BMI was also used to categorise patients according to their weight status into normal weight (BMI 18.5–25 kg/m^2^), overweight (BMI 25–30 kg/m^2^) or obese (BMI > 30 kg/m^2^). Waist circumference was measured to the nearest 0.1 cm using a non-elastic tape and with the patient standing, at the end of a gentle expiration. The measuring tape was placed around the trunk, midway between the lower rib margin and the iliac crest.

### 2.8. Rating Hunger, Fullness and Palatability 

Qualitative data on hunger, fullness, palatability and visual appeal were recorded for each study participant by the dietitian before breakfast, lunch (i.e., after morning tea) and dinner (i.e., after afternoon tea) over the 3 days using a version of Visual Analogue Scale (VAS) adapted and previously validated for use in post-prandial single meal studies by Flint et al. [21]. To rate hunger and fullness, each participant was asked questions with answers ranging from not at all to very much (i.e., 1 = not at all, 2 = somewhat, 3 = neutral, 4 = mostly and 5 = very). To rate palatability and visual appeal, each participant was asked questions with answers ranging from good to bad (i.e., 1 = good, 2 = neutral, 3 = bad). 

### 2.9. Bowel Function

At the end of each day of the study, participants reported on stool consistency to the dietitian using the Bristol Stool Scale, which provides a visual chart that classifies defaecations based on a 7-point stool hardness scale (1, hard; 7, watery) and is used to define constipation; diarrhoea and ideal-stool defaecations. This medical aid is currently the most popular diagnostic tool used in many clinical trials to assess stool consistency [22].

### 2.10. Continuous Glucose Monitoring

Continuous Glucose Monitoring (CGM) technology was employed in this study to more accurately detect glucose level fluctuations during the 3-day study compared to finger prick glucometers [23,24]. The CGM devices were calibrated against an average of four finger prick spot tests per day to ensure blood glucose and interstitial glucose measurements were comparable. During the study, subcutaneous interstitial glucose levels were recorded in 5 min intervals, providing 280 readings per day. Patients were fitted with the CGM device on the evening prior to day 1 of the study. The CGM sensor was inserted just below the skin on the subject’s abdomen by the dietitian (SS), in scope of practice as the hospital’s Accredited Practising Dietitian and Credentialed Diabetes Educator. Other CGM monitoring equipment utilised were the Medtronics iPro2 Digital Recorder, MMT-7741 (iPro2) and iPro2 Docking Station, MMT-7742 (Dock) (Medtronics, Northridge, CA, USA). At completion of the protocol (after dinner on day 3), the sensor was removed from the abdomen of the participant, and the collected data from the CGM monitor were uploaded onto the Medtronic CareLink iPro Therapy Management Software for Diabetes database (CareLink iPro, MMT-7340, Northridge, CA, USA).

### 2.11. Statistical Analysis

All data are reported as mean (standard deviation: SD or standard error: SE) or as frequencies and percentages (%). The chi-square test was used to assess the differences observed in hunger, fullness, palatability and bowel function scores among the three groups of participants. The Kolmogorov–Smirnov test was used to determine normality of distribution of the examined continuous variables. Differences with regards to the number of biscuits consumed at each mid-meal were examined with the use of the non-parametric Kruskal–Wallis test. In addition, repeated measures analysis of variance (RM-ANOVA) was used to evaluate the significance of the differences between groups in mean interstitial glucose levels, at pre-prandial and at 30, 45, 60, 75 and 90 min post-prandial (treatment effect), the significance of the changes in interstitial glucose levels observed within each group from pre-prandial to all post-prandial time points (time effect) and the differences between groups in the changes observed in interstitial glucose levels from pre-prandial to 90 min postprandial (treatment x time interaction effect). All statistical analyses were controlled for amount of carbohydrates consumed at each meal, in order to compensate for the confounding effect that could be caused by the imbalance of carbohydrate intake on CGM levels. All P-values reported are two-tailed. Statistical analysis was conducted with SPSS (version 24.0). The level of statistical significance was set at *p* < 0.05.

## 3. Results

### 3.1. Participant Study Flow and Baseline Characteristics of the Pilot Study Cohort

From the total number of 25 patients initially recruited to this pilot study, 4 patients dropped out due to early discharge from hospital, and 1 patient dislodged their CGM, resulting in an incomplete data set. Figure 1 illustrates the flow of participants in the study. 

The baseline anthropometric, demographic and medical history characteristics of the 20 participants who commenced the study are reported in Table 2. 

Overall, the participants represented older-aged adults, mean age 74 ± 12 years, mostly males (60%), and 70% were Australian-born. Reflective of this patient cohort (i.e., T2DM), 25% of participants were overweight; 60% were obese with a mean BMI of 31 ± 12 kg/m^2^ and had a mean waist circumference of 92 ± 14 cm. Mean blood glucose levels at baseline, recorded the day before CGM initiation, was 8.7 ± 2.0 mmol/L. Participants were taking various medications for their T2DM condition prior to and post-surgery, such as biguanides (55%), dipeptiyl-peptidase-4 (DPP-4) inhibitors (5%), sulphonylureas (5%) and, often, combined treatment with more than one diabetes medication, mainly biguanides combined with sulphonylureas (35%). Participants were in hospital for the following categories of surgical procedures: cardiac (45%) orthopaedic (30%), gastrointestinal (20%) and urogenital (5%). On average, participants were recruited into the study on days 2 or 3 post-operatively, depending on their respective surgeon’s discretion in approving patient transition from a post-operative clear fluids and free fluids diet onto a light ward diet.

### 3.2. Satiety and Palatability and Visual Appeal Properties of the Biscuit Treatments 

The self-reported sense of satiety (i.e., senses of hunger and fullness), palatability and visual appeal of the biscuits, as well as bowel function after consuming the different biscuits are summarised in Table 3. A higher percentage of study participants reported that when they consumed the lupin biscuits, compared to the spelt and Arnott’s biscuits, they felt a stronger sense of fullness following the afternoon tea mid-meal and before dinner (36.8% vs. 5.3% and 5.6%, respectively, *p* = 0.018). A higher percentage of study participants reported they preferred the visual appeal of the Arnott’s biscuits compared to the spelt and lupin biscuits (77.8% vs. 15.8% and 26.3%, respectively, *p* < 0.001) and that the Arnott’s biscuits were more palatable compared to these two other biscuit types (83.3% vs. 0% and 21.1%, respectively, *p* < 0.001) (Table 3). No significant differences in consistency of faeces (constipation, normal and loose bowels) were observed following any of the treatment arms (Table 3). 

### 3.3. Adherence to Biscuit Regime and Total Carbohydrate Intake

Table 4 presents the adherence of study participants to treatment (i.e., portion of biscuits consumed in mid-meals) and the total amount of carbohydrates consumed per main and mid-meal, as well as the differences among treatments. Regarding adherence, the study participants consumed a greater portion of the lupin biscuits on day 1 (27g out of 30g) compared to the portion of spelt (18g out of 30g) and Arnott’s biscuits (7g out of 16g) consumed on days 2 and 3, respectively (*p* < 0.001). Similarly, the dietary intake of carbohydrates per main and mid-meal was found to be higher for study participants on day 1 (i.e., when lupin biscuits were offered), compared to days 2 and 3 (when spelt and Arnott’s biscuits, respectively, were offered) (*p* < 0.001) (Table 4). Participants also consumed less food overall on days 2 and 3, compared to day 1 of the study.

### 3.4. Effect of Biscuits on Interstitial Glucose Levels

Figure 2 summarises the changes in interstitial glucose levels observed from pre-prandial to 90 min postprandial at breakfast, morning tea, lunch, afternoon tea and dinner after also controlling for the dietary intake of carbohydrates per meal. During breakfast, morning tea, lunch and afternoon tea, there were no significant differences among the three treatment arms. After dinner, however, on days that participants consumed the spelt and Arnott’s biscuits at morning and afternoon tea, significant increases in interstitial glucose levels, by 1.4 mmol/L (95% CI. 0.4 to 2.4) and 2.0 mmol/L (95% CI. 1.0 to 3.1), respectively, were recorded (Figure 2). In contrast, following consumption of the lupin biscuits, study participants had significantly lower interstitial glucose levels at dinner time (by −0.2 mmol/L, 95% CI. −1.2 to 0.9; *p* < 0.001) (Figure 2), suggesting a possible second meal effect.

## 4. Discussion

The key findings of this pilot pragmatic intervention study in post-surgery hospital inpatients with T2DM, were improved satiety and a significant decrease in interstitial glucose levels after dinner in those participants who consumed lupin biscuits for their mid-meal snacks at morning and afternoon tea. This phenomenon, known as the second-meal effect, is indicative of a favourable post-prandial glucose response at a subsequent meal, usually caused by a longer lasting glycaemic control produced by the consumption of legumes or wholegrains [19]. In this context, previous studies [19,25,26] have noted the positive impact of legume consumption on glucose control and insulin sensitivity, either as an immediate or as a subsequent meal effect; however, no study so far has examined the second-meal effect of lupin.

Although lupin has not been widely studied, preliminary data in non-diabetics suggested beneficial effects on satiety, blood pressure and bowel health due to its prebiotic fibres and fermentable sugars that can promote the growth of bifidobacteria (see review [17]). Hall et al. [27] previously reported a significant reduction in the postprandial glycaemic response in non-diabetic participants that consumed white bread with Australian sweet lupin flour added, compared to those that consumed standard white bread alone. Nevertheless, another study did not report any significant difference on postprandial glucose levels between overweight non-diabetics that consumed bread made with lupin kernel flour and those that consumed white bread [28]. The number of studies which have investigated the glycaemic effect of lupin in patients with T2DM are equally scarce. In this regard, only two human studies to date have demonstrated a glucose-lowering effect of lupin in adults with diabetes [29,30]. The implication of this observation is that more research is needed in the examination of the effect of lupin on glycaemic response and especially the second-meal effect in patients with T2DM. 

To the best of our knowledge, this is the first study to examine the effects of consuming lupin-enriched biscuits on the immediate or subsequent post-prandial glycaemic response in post-surgical patients with T2DM. Although it would be ideal to have the same energy content and nutrient composition in all biscuits, this was not feasible and represents a common methodological issue when examining the effects of different types of foods on specific outcomes. The approach usually followed in food intervention trials is to adjust the amount of food provided to study participants, as a means to harmonise the different interventions for those nutrients that have a main and/or direct effect on the examined outcome(s) [31]. In the case of the present study, these nutrients were carbohydrates and (to an extent) sugars, since these have the most pronounced effect on postprandial interstitial glucose levels, compared to other nutrients. Furthermore, the occurrence of the second-meal effect after dinner, but not lunch, indicates that the effect of lupin on glycaemic response is cumulative and requires some specific time to manifest. Based on our findings, it appears that two lupin biscuits consumed at morning tea, providing the equivalent of one-half tablespoon of lupin flour, are not adequate to induce a glycaemic response at lunch. However, when two additional biscuits were consumed later in the day at afternoon tea, the total dose of one tablespoon of lupin flour seems to be adequate in inducing a favourable postprandial glycaemic response at dinner. In addition, there are studies showing that colonic fermentation of legumes produces short-chain fatty acids, such as propionate, which is absorbed from the colon to the blood, slowing down the rate of gastric emptying and stimulating insulin production from the pancreas [32,33,34]. These changes, particularly the slowing of gastric emptying, affects glycaemic response several hours after legume consumption, which in the case of our study could provide an explanation for the second-meal effect observed only after dinner.

Although the nutrient profile of lupin could account for the observed second-meal effect, attention must also be paid to the overall quantity of carbohydrates consumed by the patients with diabetes throughout the day. For example, our findings could also be influenced by the combined effect of lupin with carbohydrate restriction due to partial consumption of main meals in these patients. However, the spelt biscuit, with an almost identical macronutrient profile to the lupin biscuit, did not cause a second-meal effect after dinner. Furthermore, although the participants consuming a larger number of lupin biscuits showed decreased interstitial glucose values and increased satiety, the participants consuming a smaller number of Arnott’s biscuits on day 3 showed a significant increase in interstitial glucose compared to lupin and spelt. Interestingly, this was in spite of less food being consumed on day 3. The increase in interstitial glucose may reflect the high GI nature of the Arnott’s biscuits which can impact blood glucose levels despite eating less food. In contrast, on the lupin biscuit day, the study subjects consumed more biscuits and more of their meals (including more carbohydrate) without negatively impacting blood glucose levels. 

The total quantity of lupin flour in the four lupin biscuits was about one tablespoon. It is noteworthy that this small amount of lupin flour had a beneficial impact on post-prandial glycaemia against a moderate daily intake of carbohydrate (~140 g), indicating a favourable effect on glycemic control. In addition to the low GI, high protein, high fibre and prebiotic content of lupin biscuits, this favourable effect on glycaemia may also be attributed to other bioactive nutrients in lupin, such as the glycoprotein γ-conglutin which has been associated with lower blood glucose levels [17]. In this regard, there is evidence demonstrating that diets combining lower carbohydrate intake (less than 130 g/day) with the consumption of foods that have a low GI and are high in fibre and protein can positively impact prevention and management of T2DM [11,20,35]. Despite this evidence, the Agency for Clinical Innovation (New South Wales, Australia) is recommending “moderate” carbohydrate diets within hospitals providing more than 130 g/day (30–75 g of carbohydrate for main meals and 15–30 g carbohydrate for mid meals) [9]. However, the currently available hospital meals and snacks served to inpatients with T2DM are not necessarily compliant with the recommended ‘carbohydrate modified’ meals and include options that have a high GI, low fibre and protein content. In the absence of tight regulation, hospital diets for patients with T2DM may need to become more prescriptive in the amount and type of carbohydrates to facilitate euglycaemia. In this regard, the current study implemented the moderate carbohydrate diet recommendations of the Agency for Clinical Innovation, rather than a low-carbohydrate diet which is not standard practice currently in Australian hospitals. Subjects were therefore provided with a total of 155 g of carbohydrates per day via three main meals (45 g of carbohydrate each) and two mid-meal snacks (10 g of carbohydrates each). 

Although total carbohydrate consumption approached the total amount of carbohydrates provided to study participants through their meals on the first day, this was not the case on the second and third days, when the majority consumed less than 100 g of carbohydrates. Paradoxically, although the amount of carbohydrate consumed in the last two days of the study was reduced and constituted a low carbohydrate intake, the glycaemic response was more favourable on the higher-carbohydrate diet consumed on the first day when lupin biscuits were administered. Future implications from this observation suggest that replacing the Arnott’s biscuit with the lupin biscuits at mid-meals can achieve both a lowering of the carbohydrate load and GI, with the additional benefit of the second-meal effect as demonstrated in this study. Furthermore, taking into consideration that the Arnott’s biscuit serving had to be halved to become carbohydrate matched with the lupin and spelt biscuits, replacing the Arnotts’s biscuit with lupin biscuit as mid-meal snacks would not only lower the total carbohydrate amount, but also halve the total sugars and saturated fats otherwise regularly offered and consumed in the hospital setting. 

The findings of the present study should be interpreted in light of its strengths and limitations. Regarding strengths, all statistical analyses conducted to test the research hypothesis controlled for the amount of total carbohydrate consumed so as to compensate for the potential confounding effect that could be caused by the imbalance of carbohydrate intake on CGM levels. However, it was not ethically appropriate to tightly curtail menu options in order to control for actual total food consumption including an inability to control the amount of food consumed or food rejected by an unwell inpatient population. Another strength of the present study was the use of CGM monitors to assess glycaemic response, since this allowed for detection of dynamic changes in post-prandial glucose concentrations, including post prandial transient fluctuations, which represent readings that are often difficult or impossible to collect with the use of the self-blood glucose monitoring finger-prick tests [23]. Lastly, the study was conducted within a hospital setting, which was as close as possible to controlled inpatient conditions. As far as limitations were concerned, the absence of a wash-out period between interventions in the study design may have impacted the study results. However, due to practical time constraints under which this pilot pragmatic intervention study was implemented, mainly with regards to the average length of stay of these patients in the hospital that did not exceed 3 to 4 days, the addition of a wash-out period between the different interventions was not feasible. Another limitation possibly affecting our findings could be the order with which the different types of biscuits were administered to study participants, since patients should have ideally been randomly allocated to the order of the biscuits. However, the order was decided based on the biscuits’ nutrient profile and glycaemic index, by first administering the biscuit type that would have the smallest carry-over effect on interstitial glucose levels from one day to the next. Furthermore, although the study was conducted in a relatively controlled hospital setting, post-surgery participants included patients undergoing a wide variety of surgical interventions (including total knee replacements, cardiac stents, gastrectomies, amputations, bowel resections and laminectomies), which could represent potential confounders to our results considering the varying complexity of each procedure and the effect of different types of surgery on glycaemic control [6,36,37,38,39]. Given the above, all statistical analyses were repeated after excluding the four patients enrolled in the study following gastrointestinal operations. However, there were no differences compared to the original results observed on the total sample. Also, due to the nature of post-operative studies in hospital settings, these patients are often unwell and may not consume full meals. It was noted that participants were more motivated to adhere to the intervention on the first day of the study, compared to the second and third day, possibly associated with a heightened incentive and commitment at the commencement of the trial. This may have resulted in a higher consumption of lupin biscuits on the first day, compared to the consumption of spelt and Arnott’s biscuits as the study progressed into the second and third day. 

Participants felt less hungry/fuller consuming the lupin biscuits, with increased satiety most likely associated with lupin’s higher protein and fibre content. Although participants reported feeling hungrier/less full on days 2 and 3 compared to day 1 of the study, instead of consuming more, they consumed less food compared to day 1 of the study. Paradoxically, although they favoured the more familiar visual appeal and taste of the standard hospital Arnott’s biscuit, they consumed much smaller amounts by day 3 of the study. Possible reasons explaining decreased food consumption on days 2 and 3 of the study could be a consequence of cumulative psychological and taste fatigue associated with hospital food and hospital environment as a result of conforming to a set daily mid-meal regime, the gradual loss of novelty for compliance to the trial over the three-day period post-surgery, recommencement of polypharmacy (related to nausea, hunger, satiety), as well as less passive eating in bed due to progressive increased mobility via increased mandated physiotherapy sessions. Lastly, although prescribed oral hypoglycaemic medications were recorded, they were not considered when evaluating the results. However, the fact that all study participants also served as their own controls minimised the potential confounding effect of the aforementioned limitation, since the study sample was homogenous during the three days of treatment.

Despite the limitations, this study indicates possible applications in informing revision of guidelines to optimizing therapeutic hospital meals and snacks. Specifically, it addresses the current hospital high-GI, low-fibre and low-protein, nutrient-poor mid meal snacks commonly offered to the T2DM patient population [12,40]. Due to lupin’s superior composition of protein and fibre, prebiotics and the markedly reduced carbohydrate content, biscuits made from lupin flour could be a particularly good choice for menu inclusion for inpatients with T2DM [17]. 

## 5. Conclusions

In conclusion, the present findings of a significant second-meal metabolic effect of lupin biscuits on interstitial glucose concentrations in post-operative hospital inpatients with T2DM provide some initial indications on the beneficial physiological role of this legume on glycaemic control. Our findings suggest that as little as one tablespoon of lupin flour a day may help glycaemic control and appetite in T2DM, which can have important implications for both free-living and hospitalised T2DM patients. Furthermore, our findings also suggest that people with diabetes may be able to consume a moderately higher carbohydrate intake (greater than 130 g/day) if the diet includes lupin-enriched foods. All these also add to the weight of evidence for revision of current Australian hospital menu standards in order to include mid-meal options having low GI, high fibre and high protein to achieve better post-operative glycaemic control and reduced hospital length of stay. Replacing the current commercial variety Arnott’s biscuit choice with the lupin biscuits at mid-meals could both lower total carbohydrate intake and GI, thereby reducing glycaemic load, and have the additional benefit of the second-meal effect resulting in a significant difference to glycaemic control in these patients. However, the present study represents a pilot pragmatic intervention implemented in a real-life setting; thus, it is subjected to practical constraints and limitations. Also, the exact mechanism for lupin-induced lowering of postprandial glucose concentrations remains unclear. Therefore, future larger and better controlled clinical trials with lupin biscuits are required to confirm the second-meal effect and the underlying mechanism in people with T2DM.

## Figures and Tables

**Figure 1 nutrients-12-01239-f001:**
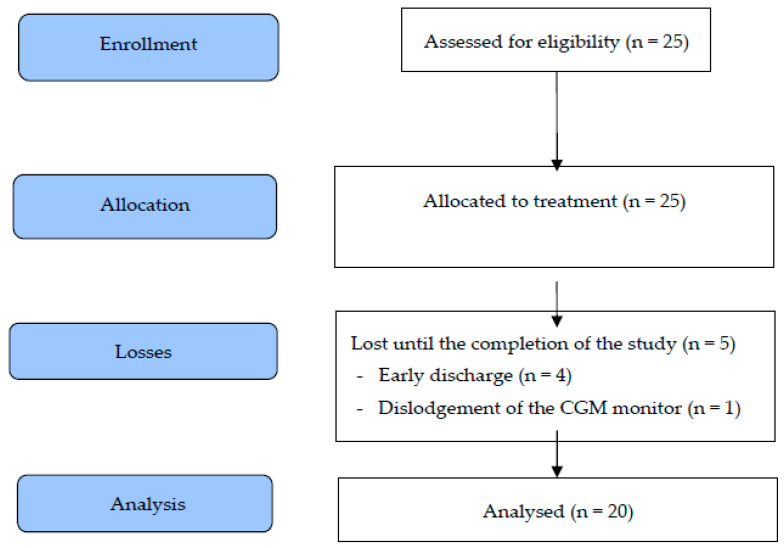
Flow diagram of study participants. CGM, continuous glucose monitoring.

**Figure 2 nutrients-12-01239-f002:**
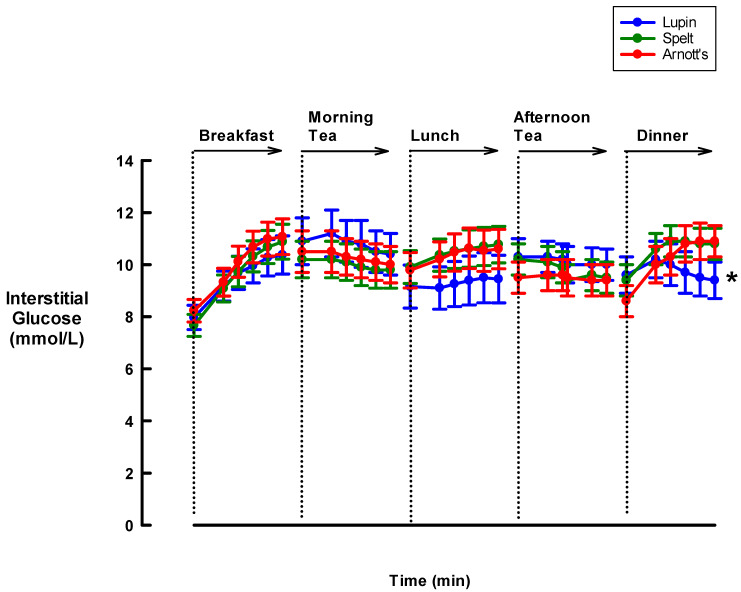
Effect of three different biscuits on interstitial glucose levels of Type-2 Diabetes Mellitus (T2DM) hospital inpatients during mid-meals and main meals. The figure presents results from repeated measures ANOVA reporting the between-group differences in mean interstitial glucose levels at each time point (treatment effect), the within-group changes in interstitial glucose levels from pre-prandial to all post-prandial time points (time effect) and the between-group differences in the changes from time (T)-0 to T-90 min (treatment x time interaction effect) at breakfast, morning tea, lunch, afternoon tea and dinner. All statistical analyses were controlled for the amount of carbohydrates consumed at each meal. The figure presents mean values (dots) and relevant standard errors of the means (whiskers). (* *p* < 0.001 in the comparison of the differences in interstitial glucose changes from pre-prandial to 90 min post-prandial, among the three treatments arms.)

**Table 1 nutrients-12-01239-t001:** Nutrient composition of lupin, spelt and standard hospital mid-meal biscuits.

Nutrient Content	Lupin Biscuits (100 g) (Skinnybik™)		Spelt Biscuits (100 g) (Skinnybik™)		Marie Biscuits (100 g) * (Arnott’s™)	
	100 g	30 g (2 Biscuits)	100 g	30 g (2 Biscuits)	100 g	16 g (2 Biscuits)
Energy (kJ)	1590	477	1590	477	1850	277
Protein (g)	12.9	3.9	10.0	3.0	6.6	1.0
Total fat (g)	14.9	4.4	15.4	4.6	11.9	1.9
Saturated fat (g)	1.6	0.5	1.8	0.6	5.9	0.9
Total carbohydrate (g)	36.0	10.8	36.0	10.8	75.0	12.0
Sugar (g)	14.9	4.5	14.0	4.2	21.3	3.4
Fibre (g)	10.0	3.0	10.0	3.0	2.8	0.4
Sodium (mg)	200	60	114	34	192	30

* Marie (Arnott’s) biscuits are the standard mid-meal biscuits served to patients at Warringal Private Hospital.

**Table 2 nutrients-12-01239-t002:** Baseline characteristics of the study population.

Descriptor	Total Sample (*n* = 20)
	Mean (SD)
Age (years)	74.3 (11.7)
Height (cm)	164.4 (7.4)
Weight (kg)	82.5 (15.6)
BMI (kg/m^2^)	30.7 (4.5)
Gender	N (%)
Female	8 (40)
Male	12 (60)
Ethnicity	
Australian	14 (70)
Non-Australian	6 (30)
Medication	
Combined Hypoglycaemic Treatment	7 (35)
Biguanides	11 (55)
DPP-4 Inhibitors	1 (5)
Sulphonylureas	1 (5)
Categories of Surgery	
Orthopaedic	6 (30)
Gastrointestinal	4 (20)
Cardiac	9 (45)
Urogenital	1 (5)
Weight Status	
Normal weight (BMI, 18.5–25 kg/m^2^)	3 (15)
Overweight BMI, 25–30 kg/m^2^)	5 (25)
Obese (BMI > 30 kg/m^2^)	12 (60)

BMI, body mass index; DPP4, dipepdidyl-peptidase-4.

**Table 3 nutrients-12-01239-t003:** Differences in satiety, palatability and bowel function among patients consuming different types of biscuits.

	Lupin Skinnybik™ (*n* = 19)	Spelt Skinnybik™ (*n* = 19)	Marie Arnott’s™ (*n* = 18)	
Satiety	n (%)	n (%)	n (%)	*p*-Value ^†^
**Hunger (“How hungry do you feel?”)**				
Before Breakfast				
Not at all/somewhat	13 (68.1)	15 (78.9)	13 (72.2)	0.745
Neutral	3 (15.8)	3 (15.8)	4 (22.2)	
Mostly	3 (15.8)	1 (5.3)	1 (5.6)	
Before Lunch				
Not at all/somewhat	15 (78.9)	11 (57.9)	15 (83.3)	0.444
Neutral	3 (15.8)	5 (26.3)	2 (11.1)	
Mostly	1 (5.2)	3 (15.8)	1 (5.6)	
Before Dinner				
Not at all/somewhat	15 (78.9)	15 (78.9)	13 (72.2)	0.249
Neutral	2 (10.5)	4 (21.0)	5 (27.8)	
Mostly	2 (10.5)	0 (0.0)	0 (0.0)	
**Fullness (“How full do you feel?”)**				
Before Breakfast				
Not at all/somewhat	9 (47.4)	12 (63.2)	9 (50.0)	0.451
Neutral	6 (31.6)	4 (21.1)	8 (44.4)	
Mostly	4 (21.1)	3 (15.8)	1 (5.6)	
Before Lunch				
Not at all/somewhat	11 (57.9)	10 (52.6)	8 (44.4)	0.155
Neutral	3 (15.8)	7 (36.8)	9 (50.0)	
Mostly	5 (26.3)	2 (10.5)	1 (5.6)	
Before Dinner				
Not at all/somewhat	10 (52.6)	10 (52.6)	9 (50)	0.018
Neutral	2 (10.5)	8 (42.1)	8 (44.4)	
Mostly	7 (36.8)	1 (5.3)	1 (5.6)	
Visual appeal of biscuits				
Good	5 (26.3)	3 (15.8)	14 (77.8)	0.001
Neutral	14 (73.7)	15 (78.9)	4 (22.2)	
Bad	0 (0.0)	1 (5.3)	0 (0.0)	
Palatability of biscuits				
Good	4 (21.1)	0 (0.0)	15 (83.3)	0.001
Neutral	13 (68.4)	19 (100.0)	3 (16.7)	
Bad	2 (10.5)	0 (0.0)	0 (0.0)	
**Bowel Function (Based on Bristol Stool Chart)**				
Constipation	5 (26.3)	3 (15)	3 (15.8)	0.777
Normal	14 (73.7)	15 (78.9)	15 (78.9)	
Loose Bowels	0 (0.0)	1 (5.3)	1 (5.3)	

^†^*p*-values were derived from chi-square tests.

**Table 4 nutrients-12-01239-t004:** Differences in the adherence of study participants to treatment (i.e., grams of biscuits consumed in mid-meals) and the total amount of carbohydrates consumed per main or mid meal.

	Lupin Skinnybik™ * (*n* = 20)	Spelt Skinnybik™ * (*n* = 20)	Marie Arnott’s™ * (*n* = 20)	
	Mean (SD)	Mean (SD)	Mean (SD)	*p*-Value ^†^
**Adherence (Grams of Biscuit Consumed)**				
Morning Tea	27.1 (3.1)	17.2 (3.5)	7.0 (2.4)	<0.001
Afternoon Tea	27.1 (3.1)	19.7 (3.4)	7.5 (2.5)	<0.001
**Total Amount of Carbohydrate Consumed (Grams per Main Meal or Mid-Meal** **)**				
Breakfast	41.2 (6.7)	30.2 (9.8)	28.5 (10.8)	<0.001
Morning Tea	9.3 (1.1)	6.1 (2.7)	5.2 (1.8)	<0.001
Lunch	39.2 (9.5)	21.8 (4.9)	19.2 (7.1)	<0.001
Afternoon Tea	9.3 (1.1)	7.1 (5.6)	5.6 (4.1)	<0.001
Dinner	39.2 (8.1)	25.0 (10.1)	22.5 (9.7)	<0.001

^†^*p*-values were derived from the Kruskal–Wallis test. * One lupin or spelt biscuit weighs 15g and 1 Marie Arnott’s biscuit weighs 8 g.

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
