# Peer review of "Effect of Lupin-Enriched Biscuits as Substitute Mid-Meal Snacks on Post-Prandial Interstitial Glucose Excursions in Post-Surgical Hospital Patients with Type 2 Diabetes"

_nutrients, 2020, doi:10.3390/nu12051239_

Round 1
Reviewer 1 Report
Manuscript nutrients-757603 entitled “Effect of lupin enriched biscuits as substitute mid- meal snacks on post-prandial interstitial glucose excursions in post-surgical hospital patients with type 2 diabetes” reports that consumption of lupin biscuits as mid-meal snacks at morning and afternoon tea improves satiety and significantly decreases interstitial glucose levels after dinner in post-surgery hospital in patients with T2DM. There are no uniform guidelines for optimising glycaemic control in T2DM patients undergoing surgery in the pre-and per-operative phases of care and this work explores how hospital meals and snacks could play and important role in managing dysglycaemia in surgical patients with T2DM. Previous studies by the same research group have suggested that Australian sweet lupins can lower blood glucose and have anti-inflammatory actions (Kouris-Blazos & Belski, 2016).
Major comments:
- The findings presented herein, although interesting, are not conclusive due to study design details. A cross-over design is appealing in this particular setting but the authors have to consider carry-over effects when submitting patients to the different interventions without a wash-out period. Also, dietary interventions lasting 1 day (2 mid-meal snacks) are not conclusive at all and it is difficult to conclude that major findings are due to the intervention and not to other bias factors not mentioned in the study (ex: post-op complications, less analgesia and pain, inflammation, etc ). To add importance, novelty and robustness to the conclusions the intervention should have been longer, probably performed in a diabetic population not submitted to surgery, and a wash-out period should have been considered between treatments.
- Exactly how many days post- surgery did the diet intervention started? The authors refer 2-3 days post-op but it is not clear exactly how many volunteers initiated the protocol 2 and 3 days after the surgery. Why was this interval different? Was it related to the type of surgery? Did the authors had access to blood testing and to inflammatory biomarkers levels like CRP? Different acute phase response status may influence the results
- Participants consumed less food overall on day 2 and 3, compared to day 1 of the study. The authors refer that during the 3-day testing period patients recommenced polypharmacy (related to nausea, hunger, satiety) and initiated physiotherapy. These interventions are plausible to cause stress in the volunteers, leading to activation of the sympathetic response and simultaneously, to an anorexigenic effect. Sympathetic activation may cause glucose dysmetabolism and increased glucose levels, justifying the differences observed in glucose excursion profiles between the 1st and 2nd and 3rd It would be important to clarify exactly when did the physiotherapy sessions started for each patient, as well as medication reintroduction and also analgesia weaning.
- Another issue relates to the nutritional composition of the biscuits. The lupin, spelt and regular biscuits were not isocaloric. There are also significant differences in the fibre and sodium content. These peculiarities must be approached in the discussion.
- Figure 2: it would be interesting to depict the glucose excursion curves after lunch and breakfast (all main meals) instead of just dinner and mid meal snacks. How do you explain the absence of a “second meal” effect after lunch, if the volunteers had already had breakfast and a lupin snack? Do you believe that the effect of lupin is cumulative, and two snacks are required for a significant change in glucose excursion curves to be observed? Authors should also consider debating chrononutritional aspects, that may help explaining why the “second meal effect” was observed only at dinner time.
- Line 342 : The authors claim that “However, the spelt biscuit, with an almost identical macronutrient profile to the lupin biscuit, did not cause a second meal effect despite being consumed after similar carbohydrate restricted meals in this patient cohort”. However, looking at the data, this does not seem very accurate: CH ingestion at lunch and dinner almost doubled in the lupin biscuit day 39.2 (8.1) compared to 25.0 (10.1) in the spelt biscuit and 22.5 (9.7) in the normal biscuit. Can the authors comment on how this difference (p<0.001) may influence the drop observed in iGlu after dinner? Was fasting blood insulin assessed in these patients? Can you comment on the effect that these differences may cause on endogenous insulin secretion in type 2 diabetic patients?
- Line 430: In conclusion, the present findings of a significant second meal metabolic effect of lupin biscuits on interstitial glucose concentrations in post-operative hospital in-patients with T2DM highlights the beneficial physiological role of this legume on glycemic control.
I believe it is ambitious to state that lupin biscuits have a second meal effect considering the study design. Confounding factors like changes in the acute phase response, changes in drug treatment ( analgesia, polypharmacy) and even the physical therapy session initiated by the patients ( line 418) may have caused stress in the volunteers, sympathetic activation and increased hepatic glucose output, disturbing post-prandial glucose control, particularly after dinner.
Reviewer 2 Report
This is an interesting clinical study with an appropriate design applied to an inappropriate study population though. Major concerns:
- Post surgical patients, especially when including patients having undergone gastric surgery is by definition an inappropriate sample as intestinal motility may affect glucose absorption and glucose levels. Therefore I suggest that these patients are excluded from the analysis performed.
- Moreover, there is are no data on specific anti diabetic treatment, as some treatments are withheld and replaced with insulin during perioperative days and these alterations may have an effect on the timing of glucose difference suggested by the authors.
- Finally the second meal effect is not obvious after the morning lupin consumption, therefore this may not be the only explanation for the observation.
Reviewer 3 Report
Thank you for the opportunity to review this study. The paper deals with a frequent problem and tries to address a common problem with an easy to perform intervention. Therefore, the paper is relevant and of interest. Furthermore, since this is a pilot study, the numbers of patients are limited and the conclusions are therefore also limited but are interesting and relevant enough to justify larger studies.
I have the following concerns with the paper:
- The main issue is the design. While I can appreciate that you performed the study using the same order of biscuits, it also introduces a bias. This is also shown by the fact that patients ate more on study day 1 than the following days. For a larger trial, I would suggest to randomly assign the order of biscuits or even better just providing one type of biscuits over the hospital course.
- The changes of biscuits every day may also introduce a bias that is not well discussed in the paper. You argue that the second meal effect is the reason why patients after the Lupin biscuit have better glycemic control in the evening. However, this may also be due to chance since the Lupin biscuit war administered the first day patients were able to start an oral diet. In a larger trial, this point must be addressed.
- The impact of the better glycemic control remains unclear. I think you need to show what type of therapy for the glycemic control was administered. How many units of insulin, which oral agents? The changes in glucose control could also be due to also be due to changes in diabetes therapy. These data must be provided.
- There should be more details on the type of surgery. It is a huge difference if a patient had a stent, an arthroplasty or an open abdominal operation such as gastrectomy regarding postoperative metabolic changes and therefore impact on glycemic control. You should stratify the outcomes depending on the surgical trauma.
- Similarly, you should provide data on complications. Usually, complications occur after day 4 and 5 and they increase also the glucose levels.
Round 2
Reviewer 1 Report
The suggested corrections to manuscript nutrients-757603 were made and all the questions raised were adequately answered by the authors.
The inaccuracies detected were corrected. The quality of writing and figures is good.The work provides interesting data regarding the effect of lupin on glycaemic response in patients with T2DM.
Reviewer 3 Report
The comments have ben well addressed